# The Effectiveness of Home-Based Inspiratory Muscle Training on Small Airway Function and Disease-Associated Symptoms in Patients with Chronic Obstructive Pulmonary Disease

**DOI:** 10.3390/healthcare11162310

**Published:** 2023-08-16

**Authors:** Wen Chang, Horng-Chyuan Lin, Hsueh-Erh Liu, Chin-Yen Han, Po-Jui Chang

**Affiliations:** 1Department of Nursing, Chang Gung University of Science and Technology, Taoyuan 33303, Taiwan; changwen@mail.cgust.edu.tw (W.C.); cyhan@mail.cgust.edu.tw (C.-Y.H.); 2Department of Thoracic Medicine, Chang Gung Memorial Hospital Linkou Main Branch, Taoyuan 33303, Taiwan; lin53424@gmail.com; 3College of Medicine, Chang Gung University, Taoyuan 33303, Taiwan; 4School of Nursing, College of Medicine, Chang Gung University, Taoyuan 33303, Taiwan; sarah@mail.cgu.edu.tw; 5New Taipei Municipal TuCheng Hospital, Chang Gung Medical Foundation, New Taipei City 236017, Taiwan

**Keywords:** chronic obstructive pulmonary disease, home-based pulmonary rehabilitation, inspiratory muscle training, small airway function, dyspnea

## Abstract

Chronic obstructive pulmonary disease (COPD) is characterized by persistent airflow limitations, occurring mainly in the small airways. Weakness in the respiratory muscles contributes to dyspnea and a decreased exercise capacity in COPD patients. This study aimed to investigate the effectiveness of home-based inspiratory muscle training (IMT) on small airway function and symptoms in COPD patients. This research adopted a non-randomized controlled-study quasi-experimental design. The IMT program consisted of two 15 min sessions·d^−1^, 5 d·wk^−1^, with 40% of the maximal inspiratory pressure (PI_max_) on each participant’s assessment results and lasted for 12 weeks. Small airway function was assessed using plethysmography at baseline and after 12 weeks. The modified British Medical Research Council (mMRC), COPD assessment test (CAT), PI_max_, and 6 min walking distance (6MWD) were recorded at baseline as well as four, eight, and twelve weeks. Twenty-three participants with at least moderate COPD were enrolled in IMT (*n* = 16) or in the control group (*n* = 7) in this study. The study participants were mostly male (82.6%), and the average age was 68.29 ± 10.87 years, with a mean body mass index (BMI) of 23.54 ± 4.79. After 12 weeks, the ratios of the first second of forced expiration to the forced vital capacity (FEV_1_/FVC%) (*B* coefficient [95% Wald confidence interval] of 5.21 [0.46 to 9.96], *p* = 0.032), forced expiratory flow (FEF_25–75%_) (0.20 [0.04 to 0.35] L/s, *p* = 0.012), and FEF_50%_ (0.26 [0.08 to 0.43] L/s, *p* = 0.004) in the IMT group were significantly better than in the control group. The IMT group showed significantly lower CAT scores at week 8 (−5.50 [−10.31 to −0.695] scores, *p* = 0.025) than the control group. The mMRC grade, CAT score, PI_max_, and 6MWD were significantly improved compared to their values at baseline in the IMT group. Home-based IMT effectively improved post-bronchodilator small airway function and disease-associated symptoms in COPD patients.

## 1. Introduction

Chronic obstructive pulmonary disease (COPD) is a serious global public health issue. It is one of the top three leading causes of deaths worldwide with a high rate of mortality and morbidity and with impacts on patients’ quality of life [1]. COPD is a chronic and progressive disease of the lung characterized by irreversible airflow limitations. The main sites of airflow obstruction in COPD are the small airways [1,2]. Spirometry has long been used clinically to evaluate the lung function of patients with respiratory diseases and to diagnose COPD [3]. There are limitations when using routine spirometry to evaluate the extensive lesions in the small airways of COPD patients because spirometry is insensitive to peripheral airway obstruction [4]. The literature indicates that small airway function offers a novel perspective on COPD, providing extensive insight into the correlations between the pathological mechanisms of COPD and lung function [5]. Body plethysmography is a sensitive lung measurement used to detect lung pathologies, and these measurements have been demonstrated to confer clinical information that might be missed by conventional pulmonary function tests, especially in obstructive airway diseases [6]. Small airway function, therefore, should be considered to be one of the essential clinical evaluation items of treatment effectiveness in patients with COPD.

The main goals of the treatment of stable COPD patients are reductions in both the symptoms and the future risk of exacerbation [1], and the management strategies include both pharmacological treatments and non-pharmacological interventions [7]. Respiratory muscle weakness, especially in the inspiratory muscles, is prevalent in patients with COPD [8,9] and contributes to dyspnea and exercise limitations [10,11]. Pulmonary rehabilitation is a comprehensive intervention aimed at improving the physical and psychological condition of patients with COPD [1]. The outcomes of pulmonary rehabilitation for patients include improvements in their exercise capacity and quality of life [12]. It is also suggested as an appropriate intervention for most patients with COPD [1].

Inspiratory muscle training (IMT) is a type of resistance training that is chosen for patients with COPD who are unable to participate in exercise training [13]. IMT is a convenient and safe exercise method that can be carried out in a home setting [9,14,15,16]. Studies have demonstrated that a standard protocol of 30% or higher (30–80%) of PI_max_ for a duration of 20 to 30 min per day for 10 to 12 weeks increases the strength in the inspiratory muscles and results in improvements in dyspnea, quality of life, and exercise performance [14,17,18]. However, the previous studies in this field have methodological heterogeneity [16], and the benefits that can be attributed to IMT in isolation and to its association with other interventions are unclear [19]. It is also unclear whether IMT improves the symptoms associated with the condition by enhancing the small airway function of COPD patients. These are challenges to applying the outcomes of IMT on COPD patients, and more research is required to clarify them.

In Taiwan, COPD is ranked eighth in the top ten leading causes of death and had an increased rate of 10.3% in 2021 [20]. COPD occurs in smokers over the age of 40, and it occurs more in men than in women; the prevalence rate is 6.1% in adults aged over 40 [21], and the prevalence rate of moderate to severe COPD is 5.4% [22] compared with the 10.3% global prevalence of COPD [1]. COPD is associated with a significant financial burden for patients, caregivers, and society as a whole. The budget spent on COPD is EUR 38.6 billion in Europe annually [1]. Home-based COPD patients with stable conditions are managed through outpatient clinic follow-ups every three months, daily drug treatments, and regular pulmonary function testing through spirometry and chest X-ray assessments without routine pulmonary rehabilitation in Taiwan. The literature indicates that the issues related to home-based care for COPD patients need to be emphasized [1]. However, little attention has been paid to the effectiveness of home-based IMT in patients with COPD in Taiwan. The aim of this study was to investigate the effectiveness of home-based IMT on small airway function, dyspnea, quality of life, exercise endurance, and inspiratory muscle improvement in patients with COPD. It was hypothesized that home-based IMT would improve small airway function and disease-related health conditions in COPD patients.

## 2. Materials and Methods

### 2.1. Study Design

This non-randomized controlled study adopted a quasi-experimental design for 12 weeks for stable COPD patients. From October 2014 to October 2017, subjects with COPD were enrolled in the pulmonary outpatient clinic of Chang Gung Memorial Hospital (Linkou Branch), where there is a level III medical center with 3700 beds, in Taiwan.

### 2.2. Participants

Participants who were determined to be eligible by thoracic medical specialists and were willing to participate were referred to the investigators. The investigators explained the study procedure, and the participants decided whether to be given home-based IMT (IMT group) or to receive standard care (control group) for 12 weeks. The participants in both groups received the same care, including regular medical treatments and assessments. The experimental group was provided with IMT, and all participants returned to the pulmonary outpatient clinic once a month for study assessments. The inclusion criteria were as follows: (1) patients with a diagnosis of COPD with post-bronchodilator forced vital capacity (FEV_1_/FVC) < 0.7 according to the 2011 Global Initiative for Chronic Obstructive Lung Disease (GOLD) criteria [1] and at least moderate airflow limitation, as patients with mild airflow limitations often had no obvious symptoms and were easily ignored without seeking medical aid [23]; (2) patients who were living at home and able to receive regular follow-ups in the outpatient clinic; (3) patients with no previous receipt of IMT; (4) patients with stable condition and no hospital stays or emergency treatments for COPD acute exacerbations in the preceding three months; (5) patients with ability to communicate in Mandarin Chinese or Taiwanese and willingness to participate; and (6) patients who were adults aged 40 years or older. The exclusion criteria were as follows: (1) comorbid left-sided heart failure, (2) acute myocardial infarction in the past three months, or (3) the requirement of partial or complete assistance in daily activities (e.g., as a result of bone fracture or cerebrovascular accidents).

### 2.3. Procedures

Based on 2011 GOLD strategy document, modified British Medical Research Council (mMRC) and COPD assessment test (CAT) were recommended as the standard for evaluating the impact of COPD symptoms on patients [1]. A 6 min walk test (6MWT), as one of the most utilized exercise tests for COPD, was used to evaluate exercise capacity and intervention outcomes [24]. Participants undertook assessments to determine their mMRC grade, CAT score, maximal inspiratory pressure (PI_max_), maximal expiratory pressure (PE_max_), and 6 min walking distance (6MWD), and they received total-body plethysmography; these data were recorded as the baseline data (T1). Data were collected again at four weeks (T2), eight weeks (T3), and twelve weeks (T4) except total-body plethysmography, which was performed at T1 and T4. This study was approved by the Institutional Review Board of the medical center (No: 103-4903B), and it received permission from the study site prior to participant enrolment. Participants were enrolled in this study only after giving written informed consent.

#### 2.3.1. The IMT Device and Prescribed Training

The selection of IMT device and training prescription was based on recommendations from the American College of Sports Medicine (ACSM) [13,25] as well as other published literature [10,14,16,18] and the safety and convenience of the participants’ home-based training [15]. The threshold IMT device (RESPIRONICS Threshold IMT, Respironics Respiratory Drug Delivery, UK, NDC NO:08373-0730-00, Medical Device permit: Taiwan Department of Health Medical Device Import Number 010248) (Figure 1) was adopted as the training device in the experimental group (IMT group). The threshold IMT device has an inspiratory threshold loading design with a spring that provides adjustable resistance (range of 9–41 cmH_2_O ± 2 cmH_2_O). The user breathes against a spring-loaded valve, and the load of the valve can be adjusted to accommodate the user’s capacity. Based on the literature [10,14,16,18], the training program consisted of 15 min sessions performed twice daily, five days a week, using 40% PI_max_ as the loading threshold and was adjusted every four weeks based on each participant’s assessment results. The training program lasted 12 weeks.

#### 2.3.2. Clinical Disease Symptom Severity Evaluations

##### The Modified British Medical Research Council (mMRC) Dyspnea Scale

The mMRC scale was modified from the British Medical Research Council dyspnea scale by the American Thoracic Society (ATS) [26]. The mMRC scale is graded from 0 to 4 and quantifies patient-reported daily breathlessness [27]. The 2011 Global Initiative for Chronic Obstructive Lung Disease (GOLD) strategy document recommends assessment of COPD using symptoms and future exacerbation risk, employing mMRC grade ≥ 2 as equivalent symptom cut points and categorizing patients into low- or high-symptom groups [28].

##### The COPD Assessment Test (CAT) Score

The COPD assessment test (CAT) score consists of eight questions, each scored 0–5 from the least to most severe impairment [29]. CAT is sensitive to COPD disease severity [30] and has been recommended by GOLD since 2011 as the standard test for evaluating the impact of COPD symptoms on patients. A CAT score of <10 indicates a less symptomatic state, while a score of ≥10 indicates a more symptomatic state [1,26].

#### 2.3.3. Measurement of Lung Function, Pulmonary Physiology, and Exercise Tests

##### Total-Body Plethysmography

The VIASYS body Box Vmax 22D model (Viasys Healthcare, Conshohocken, PA, USA) was used for taking measurements, including spirometry, lung volume, and diffusion capacity, and was operated by the study site’s lung function analysts. The study participants sat in the “body box” (body plethysmography) and closed the door to form a sealed chamber. When the door was closed, the study participants inhaled and exhaled as instructed by the analyst. By measuring the airway pressure of the participants and the change in the volume of the box, the body box calculated the participants’ lung volume through the detection of the electronic converter [31]. To illustrate lung function prior to and after the use of bronchodilators, participants were given bronchodilators upon completion of the primary tests and then took a 15 min rest before undergoing the secondary tests.

##### Manovacuometry

Manovacuometry was used to measure the respiratory muscles, including PI_max_ and PE_max_. The pressure range between inhalation (negative values) and exhalation (positive values) was ±120 cmH_2_O. The standard protocol for respiratory muscle measurements in this study is described below. Each participant was assessed individually for their PI_max_ and PE_max_ values. Prior to each measurement, manovacuometry was returned to zero. Participants were instructed to sit on a chair, to breathe naturally, and to adjust their breaths at will. After complete expiration, participants inhaled fully through a mouthpiece held in their mouths with their noses covered, and, from this, the PI_max_ was derived. In reverse, when participants exhaled fully after full inhalation, the PE_max_ was derived. Participants were asked to repeat this procedure 3 to 5 times during each assessment, and the best values were recorded for further analysis.

##### The 6 min Walking Test

This study complied with European Respiratory Society and American Thoracic Society’s walking test technical standards for chronic respiratory disease [32]. The test was performed along a test corridor at the Pulmonary Recovery Center of the study site after the test site was deemed to be qualified through a hospital evaluation of the medical center. Professional personnel in the Pulmonary Recover Center undertook the evaluation and provided verbal encouragement using specific terms every minute during the test. The participants, equipped with a pulse oximeter, were asked to walk at their fastest pace and to adjust the in-between pace on their own until the end of the 6 min test period. Spirometry was performed before and after the exercise. Investigators recorded the full walking distance achieved within 6 min (6MWD) by each participant, the extent of breathlessness before and after exercise determined by modified Borg scale (MBS) [33], and their pulse rate and oxygen saturation through the course of the exercise. The MBS, pulse rate, and oximetry measurements served as a safety monitoring tool during the exercise assessment [32,33].

### 2.4. Statistical Analysis

The primary outcome was the difference of forced expiratory flow (FEF_25–75%_) and FEF_50%_ between the IMT and the control groups at week 12. Secondary outcomes included the mMRC grade, CAT score, PI_max_, forced expiration volume in one second (FEV_1_), forced vital capacity (FVC), and 6MWD at week 12. Statistical analysis was performed using IBM SPSS statistics 19.0 software (IBM Corp., Armonk, NY, USA). Descriptive analysis was conducted for demographics and was presented as number, percentage, mean, and standard deviation as appropriate. The baseline characteristics, demographics, and comparisons of the variables in the IMT and control groups were analyzed using chi-square test for categorical variables and Mann–Whitney U test for continuous variables. Wilcoxon signed-rank test was used to analyze and compare the respective changes in the IMT group and control group during the study period. A generalized estimating equations (GEE) method is a marginal semi-parametric approach that analyzes measurements repeated over time. A GEE model was utilized to analyze and compare the study variables in the IMT group and control group in order to determine the interactions and differences between the study groups and time points (T2, T3, and T4 vs. T1) and to understand the outcomes after the IMT intervention. *p* < 0.05 (one-tailed test) was taken to indicate statistical significance.

## 3. Results

A total of 31 participants was initially enrolled in this study. During the study period, a few participants were lost to follow-up or had missing data due to an inability to adapt to procedural or environmental issues during the small airway assessment (*n* = 2), moving from Northern Taiwan due to work (*n* = 1), admittance to a hospital for an unrelated condition (*n* = 1), and non-adherence to the return visit schedule (*n* = 4). Finally, 23 participants completed the study, 16 in the IMT group and 7 in the control group (Figure 2).

### 3.1. Participant Demographics

The study participants were mostly male (82.6%), and the average age was 68.29 ± 10.87 years, with a mean body mass index of 23.54 ± 4.79. Most of the study cohort had a history of smoking (56.5%), and the majority had moderate (65.2%) or severe (30.4%) airflow limitations. The most common medications in use were long-acting β2 agonists (LABAs)/long-acting muscarinic antagonists (LAMAs), inhaled corticosteroids (ICSs)/LABAs, or ICSs/LAMAs/LABAs (26.1% each). The mean number of acute exacerbations experienced in the previous year was 1.61 ± 1.95 for the total cohort, with numerically more participants in the IMT group than in the control group having ≥2 exacerbations per year (50% vs. 14.3%, *p* = 0.176) (Table 1). No statistically significant differences in the demographic data were noted between the two groups.

### 3.2. Effects of IMT on Lung Function in COPD Participants

The study variables representing the pulmonary volume, gas diffusion capacity of the lung, and post-bronchodilator expiratory flow of the IMT and control groups at baseline (T1) and at the study endpoint (T4) are shown in Table 2. After 12 weeks of training, the expiratory flow of the VC (2.42 ± 0.73 L (T4) vs. 2.09 ± 0.47 L (T1), *p* = 0.011) and ERV (0.90 ± 0.41 L (T4) vs. 0.56 ± 0.38 L (T1), *p* = 0.012) was significantly more than at baseline in the IMT group. In the control group, the expiratory flow of the FVC (2.61 ± 0.51 L (T4) vs. 2.35 ± 0.44 L (T1), *p* = 0.043), FEV_1_/FVC% (48.00 ± 14.36% (T4) vs. 54.00 ± 11.76% (T1), *p* = 0.027), FEF_25–75%_ (0.49 ± 023 L/s (T4) vs. 0.71 ± 0.40 L/s (T1), *p* = 0.018), post-bronchodilator FEF_25–75%_ (0.48 ± 0.22 L/s (T4) vs. 0.70 ± 0.33 L/s (T1), *p* = 0.018), and FEF_50%_ (0.65 ± 0.35 L/s (T4) vs. 0.87 ± 0.51 L/s (T1), *p* = 0.018) statistically significantly changed from T1 to T4. None of the other variables of lung function (body plethysmography or spirometry) showed statistical significance regarding changes in either group.

In order to avoid bias resulting from the differences in small airway function observed between the two groups at baseline, the GEE model was used to analyze and compare the individual variables of the groups (IMT group and control group) between the study endpoint (T4) and baseline (T1). The analysis of the time points shown in Table 3 indicated that the change in the regression coefficient of FEV_1_/FVC% between T4 and T1 was statistically significant (B coefficient [95% Wald confidence interval] of −4.29 [−7.24 to −1.33], *p* = 0.004), suggesting that the total study cohort’s FEV_1_/FVC% at 12 weeks decreased by 4.29% compared with that at baseline. The interaction between the study groups and study time points showed that the regression coefficient of the interaction variable of IMT × (T4–T1) for FEV_1_/FVC% was statistically significant (B = 5.21 [0.46 to 9.96], *p* = 0.032), with FEV_1_/FVC% after 12 weeks of intervention significantly increasing by 5.21% in the IMT group compared with the control group (Figure 3). The regression coefficients for the changes in FEF_25-75%_ and FEF_50%_ between T4 and T1 were also statistically significant (FEF_25–75%_: −0.22 [−0.32 to −0.12] L/s, *p* < 0.001) (FEF_50%_: −0.22 [−0.36 to −0.09] L/s, *p* < 0.001), showing that the total study cohort’s FEF_25–75%_ and FEF_50%_ at 12 weeks decreased by 0.22 L/s compared to those at baseline. The interaction between the study groups and study time points showed that the FEF_25–75%_ and FEF_50%_ were statistically significant (FEF_25–75%_: 0.20 [0.04 to 0.35] L/s, *p* = 0.012) (FEF_50%_: 0.26 [0.08 to 0.43] L/s, *p* = 0.004), demonstrating that FEF_25–75%_ and FEF_50%_ significantly increased in the IMT group after 12 weeks of intervention compared with those in the control group (Figure 3). None of the other interaction variables showed statistical significance regarding changes in lung function.

### 3.3. Effects of IMT on Disease-Associated Symptoms in COPD Participants

The comparison of the individual variables at different time points within the IMT group and control group (Table 2) showed that the mMRC grade in the IMT group decreased over time and that, at T3 and T4, it was significantly lower than that at T1, respectively (1.00 ± 0.52 (T3) vs. 1.50 ± 0.63 (T1), *p* = 0.005; 0.88 ± 0.62 (T4) < T1, *p* = 0.008). In the control group, no statistically significant differences were noted in the mMRC scale. This result indicates that the dyspnea during daily activities in the IMT group significantly improved as the training program progressed, and the benefit was readily observable at eight weeks, with a continuous improvement throughout the intervention period. The CAT score decreased from scores of 11.50 ± 6.16 (T1) to scores of 8.88 ± 6.54 (T4) in the IMT group, and T3 was significantly lower than T1 (9.00 ± 6.07 scores (T3) < T1, *p* = 0.032); however, it increased from scores of 11.29 ± 9.01(T1) to scores of 12.57 ± 10.47 (T4) in the control group, illustrating that the impact of the symptoms on the participants’ quality of life improved in the IMT group (Figure 4), with a mean CAT score of > 10 prior to the intervention, progressing to ≤10 after 12 weeks of intervention. In contrast, the CAT scores in the control group increased over time and were >10 points at both study time points. Changes in PI_max_ (84.31 ± 23.72 cmH_2_O (T4) vs. 72.13 ± 26.75 cmH_2_O (T1), *p* = 0.011) also showed statistical significance, demonstrating that PI_max_ significantly increased after 12 weeks of training in the IMT group (Figure 4). Additionally, in the IMT group, the 6MWDs at T2 and T3 (437.63 ± 64.74 m (T2) vs. 424.44 ± 66.40 m (T1), *p* = 0.012; 443.38 ± 59.07 m (T3) > T1, *p* = 0.017) were significantly improved compared to those at baseline (Figure 4), suggesting that the benefit of enhanced exercise endurance was readily observable at four weeks, with continuous improvement at both week 4 and week 8. In the control group, no statistically significant differences were noted in PI_max_ or the 6MWD during the study period.

Although the resting heart rate (RHR) was significantly higher in the IMT group than in the control group at the study baseline, it decreased over time in the IMT group (80.13 ± 10.23 bpm (T4) vs. 87.56 ± 13.04 bpm (T1), *p* = 0.011), but the RHR significantly increased from baseline to week 12 (79.86 ± 10.42 bpm (T4) vs. 70.71 ± 12.83 bpm (T1), *p* = 0.028) in the control group (Figure 5). PE_max_ was not analyzed in the current study, as it fell outside of the detection range (>120 cmH_2_O) in some participants.

The GEE model was utilized to compare the changes in the two study groups during the study period and to understand the outcomes after the IMT intervention. As shown in Table 3, the regression coefficient of IMT × (T3–T1) for the changes in the CAT scores was statistically significant (−5.50 [−10.31 to −0.695] scores, *p* = 0.025), indicating that the impact of the disease symptoms on the participants’ quality of life further improved by 5.50 points in the IMT group compared with the control group at week 8 (Figure 4). The regression coefficient of the study group for the changes in RHR was statistically significant at baseline (*p* = 0.002), showing that RHR in the two groups was not comparable. After eliminating the non-comparability of RHR between the two groups at baseline, the change in the RHR between T3 and T4 was statistically significant (7.00 [2.37 to 11.64] bpm (T3), *p* = 0.003; 9.14 [3.82 to 14.46] bpm (T4), *p* = 0.001), suggesting that the total study cohort’s RHR at 8 weeks increased by 7 bpm and that, at 12 weeks, it increased by 9.14 bpm compared with the value at baseline. The interaction between the study groups and study time points showed the regression coefficients of the IMT × (T3–T1) and IMT × (T4–T1) interaction variables were statistically significant (IMT × (T3–T1) of −10.88 [−18.22 to −3.53] bpm, *p* = 0.004; IMT × (T4–T1) of −16.58 [−23.51 to −9.65] bpm, *p* < 0.001), indicating that the changes in RHR achieved at 8 weeks and 12 weeks in the IMT group were significantly lower than those in the control group by 10.88 bpm and 16.58 bpm, respectively (Figure 5). The 6MWD and PI_max_ were significantly increased after training in the IMT group compared to those at baseline, but none of the changes in the 6MWD and PI_max_ achieved a statistically significant difference between the two groups.

## 4. Discussion

### 4.1. Beneficial Effects of IMT on Lung Function and Heart Rate in COPD Participants

The small airway function assessments in the present study found that the alveolar gas diffusion, airway obstruction, and small airway airflow limitations in the COPD participants tended to deteriorate with time. When analyzing the interactional effects between the study groups and the study time points, post-bronchodilator FEV_1_/FVC%, FEF_25–75%_, and FEF_50%_ all showed significant improvements in the IMT group compared with the control group. Spirometry has been commonly utilized to evaluate lung function in past studies on IMT. Past studies showed variable results: Basso-Vanelli et al. [8] investigated an IMT program that involved a 21 min session each day for four consecutive months in 13 COPD patients, but they found that FEV_1_/FVC%, FEV_1_, and FVC were not significantly different after training. Mehani [34] evaluated a training strategy using 15–60% PI_max_ three times per week for two months in 20 elderly patients with moderately severe COPD. They found that FEV_1_, FVC, predicted FEV_1_%, and predicted FVC% were significantly increased from baseline after the IMT. A FEV_1_/FVC% of < 0.70 is an indicator for a COPD diagnosis, suggesting limited expiratory airflow [1]. FEF reflects the severity of the airway obstruction, and decreases in FEF_25-75%_ and FEF_50%_ may be a result of small airway obstruction [31]. Distal expiratory flows, such as the forced expiratory flow at 50% FVC (FEF_50_), are important functional parameters for diagnosing small airway disease [35]. The results of the present study suggest that IMT could ameliorate the airway obstruction progression that occurs with COPD. Body plethysmography can effectively differentiate the COPD severity, which may be a supportive method for assessing the lung function of stable COPD patients [36]. Although this study found no differences in the parameters of body plethysmography in the participants, this may be because the study period was only 12 weeks and not the required long-term training and follow-up of the changes in small airway function.

The assessment of the heart rate has been used as a marker of health [37]. Having a lower RHR is usually a sign of good health and could therefore be a marker of a healthy lifestyle [38]. A decrease in RHR after exercise training may indicate enhanced cardiopulmonary endurance [39]. Conversely, a high RHR is associated with disease and adverse events [37], and the complications associated with a high rate include lower energy levels, low physical fitness, reduced blood circulation, etc. In the present study, at 8 and 12 weeks, the participants’ RHR significantly increased from baseline in the study cohort, suggesting that physical fitness decreases with time and that small airway airflow limitations tend to lower energy levels over time, leading to an RHR increase in stable COPD patients. In contrast, the RHR of the IMT group was significantly reduced from baseline, and it especially significantly decreased compared with the control group at 8 and 12 weeks. IMT could improve exercise performance and airway obstruction, which may lead to enhanced cardiopulmonary endurance. Home-based IMT perhaps can enhance the physical health of stable COPD patients.

### 4.2. Beneficial Effects of IMT on Disease-Associated Symptoms of COPD Participants

Dyspnea is the most common symptom in patients with COPD [40]. Improving dyspnea is a major goal when treating COPD patients [14]. A systematic review and meta-analysis by Figueiredo et al. [19] found that isolated IMT for patients with COPD showed no improvement in dyspnea or their quality of life. Another systematic review and meta-analysis by Beaumont et al. [14] found that dyspnea was significantly decreased during IMT sessions, but COPD patients without a weakness in their inspiratory muscles (PI_max_ > 60 cmH_2_O) had a significant decrease in the IMT group compared with the control group. In this current study, we found that the participants (PI_max_ > 60 cmH_2_O) experienced improvements in dyspnea after eight weeks of training in the IMT group, and this beneficial effect continued throughout the study period. Meanwhile, the CAT scores were improved from being more symptomatic (≥10 points) prior to training to less symptomatic (<10 points) after training, and these changes were greater in the IMT group than in the control group. This difference is indicative of the beneficial effects of home-based IMT in improving dyspnea and symptoms in patients with COPD as well as their quality of life. The results were similar to the previous study’s results [14]. The current study found that PI_max_ and 6MWD were significantly increased after training in the IMT group, indicating enhanced exercise performance and greater inspiratory muscle strength. Peripheral muscle weakness can lead to an impaired exercise capacity in COPD patients [11]. Therefore, IMT can help COPD patients improve their inspiratory muscle strength as well as their exercise performance. To sum up the above findings, home-based IMT is beneficial for dyspnea and disease-associated symptoms in stable COPD patients, their quality of life, and their exercise capacity. These results were in concordance with the previous publications showing that IMT improved the inspiratory muscle strength and exercise endurance in COPD patients and positively impacted both their breathlessness and quality of life [9,10,14,16]. Although PI_max_ and 6MWD were significantly improved from baseline after training in the IMT group, only a trend of improvement in the IMT group compared with the control group was noted.

Chronic obstructive pulmonary disease is a heterogeneous lung disease caused by gene–environment interventions, and it occurs throughout an individual’s lifetime. COPD is characterized by chronic respiratory symptoms due to abnormalities in the airways and/or alveoli [1]. The alleviation of dyspnea symptoms is a key objective highlighted across COPD guidelines. The first step is to optimize the bronchodilator therapy to improve the respiratory mechanics and muscle function, thus increasing the exercise capacity [40]. The strength of this study was the demonstration of the beneficial effects of home-based IMT in enhancing the post-bronchodilator airflow in the small airways in association with reduced disease-associated symptoms. Moreover, the IMT device is lightweight and easy to carry, facilitating its use by COPD patients in the home setting. For COPD patients who cannot receive physical rehabilitation regularly at healthcare institutions or who cannot perform whole-body exercise endurance training, IMT is a method that may be considered and promoted for pulmonary rehabilitation.

This study’s participants were mostly male, the average age was over 65 years, and most of them had a history of smoking. These characteristics were consistent with the findings in the previous studies in which smoking, a male sex, and an older age were the characteristics of COPD patients [23]. In older adults, sarcopenia may reflect the impaired muscle quality. Cigarette smoking, low physical activity, poor diet, and older age are risk factors shared by COPD and sarcopenia that have been identified [41]. While sarcopenia may contribute to inspiratory muscle weakness, the participants in this study had an average BMI that was within the normal range. Future studies may be considered to include a nutrition assessment of the COPD patients to understand the relationship between home-based IMT, nutritional status, inspiratory muscle strength, and small airway function.

The total number of participants in this study was small, which did not meet the required sample size for estimation. Therefore, the power of this study was calculated from the results of the study. Based on the main goal of the treatment of COPD patients being to relieve the symptoms of dyspnea [1], in this study, the results of four times the mMRC dyspnea scale in the IMT group were used to calculate the power. The observed power was calculated using the repeated measures of the general linear model, and the least significant difference (LSD) was used to compare the main effect. The results showed that the test power of the mMRC was 0.957, indicating that the results of this study had a good effect size. The limitations of this study were the relatively small sample size, the characteristics of the participant demographics, the specific geographic region, and the single medical center without the generalizability of the findings to a broader population of COPD patients. More research is required in the future to document the benefits of home-based IMT regarding small airway function improvement in patients with COPD.

## 5. Conclusions

In stable COPD participants with at least moderate disease severity, 12-week home-based IMT may improve the post-bronchodilator air flow in the small airways and ameliorate disease-associated symptoms. This training modality may be a choice for home-based pulmonary rehabilitation programs.

## Figures and Tables

**Figure 1 healthcare-11-02310-f001:**
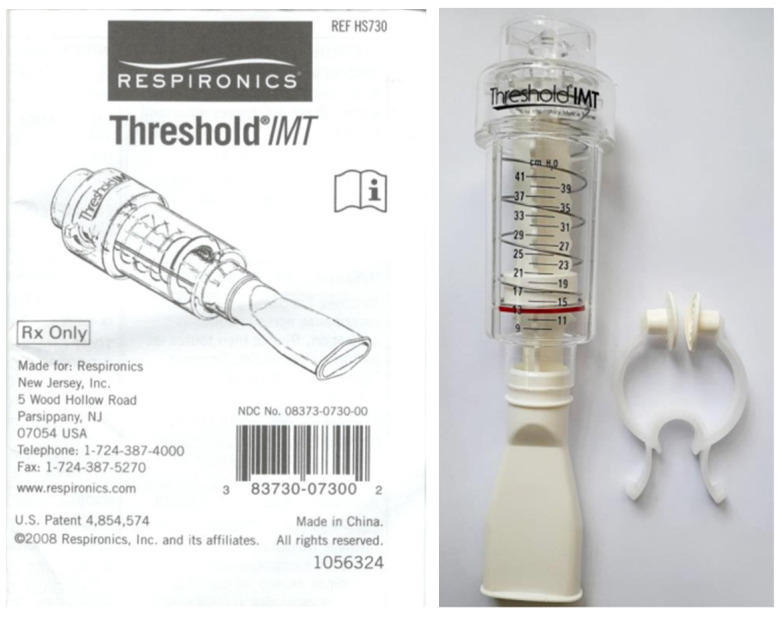
The threshold IMT device.

**Figure 2 healthcare-11-02310-f002:**
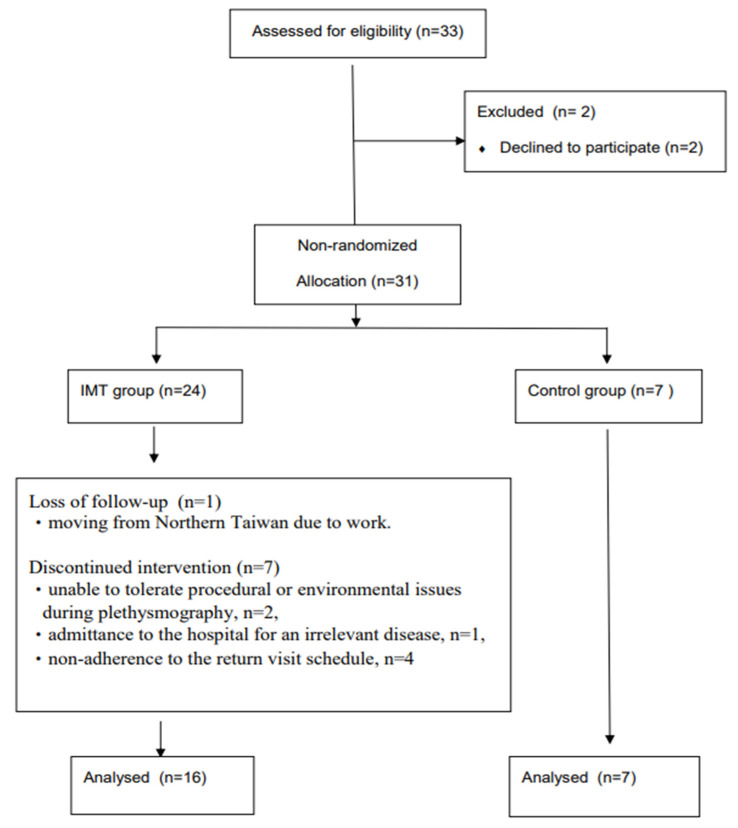
Flow diagram from enrolment to analysis process of the study.

**Figure 3 healthcare-11-02310-f003:**
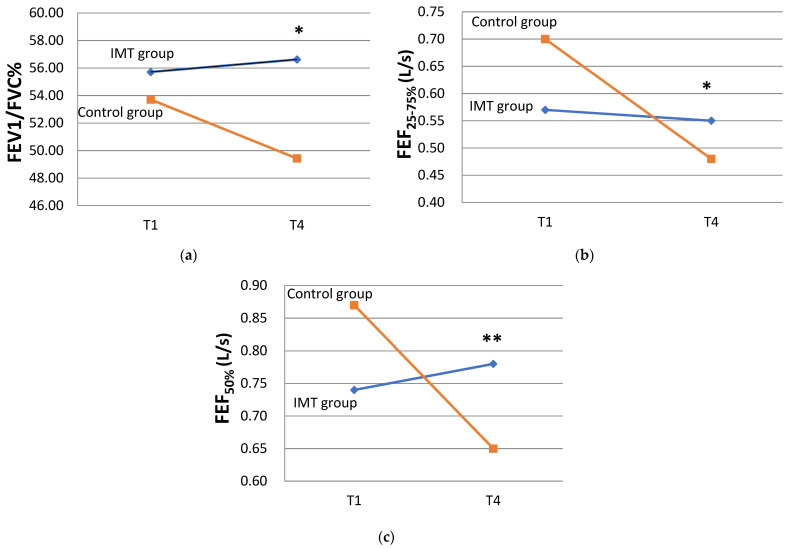
Changes in lung function from baseline to 12 weeks in the IMT and control groups. The interactions between two groups and the change from T1 (baseline) to T4 (12 weeks) were compared and analyzed through GEE. (**a**) The forced expiratory volume in 1-second-to-vital-capacity ratio (FEV1/FVC%). (**b**) Forced expiratory flow between 25 and 75% of the forced vital capacity (FEF_25–75%_). (**c**) Forced expiratory flow at 50% of the forced vital capacity (FEF_50%_). * *p* < 0.05 and ** *p* < 0.01.

**Figure 4 healthcare-11-02310-f004:**
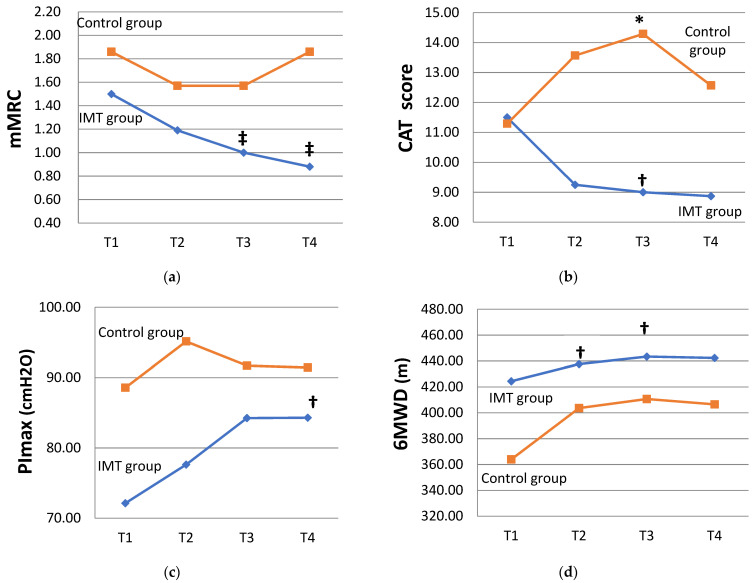
Changes in disease-associated symptoms from baseline to 12 weeks in the IMT and control groups. The interactions between two groups and the change achieved between time points were compared and analyzed through GEE. The differences in the IMT group between T2 (4 weeks), T3 (8 weeks), and T4 (12 weeks) vs. T1 (baseline) were analyzed through Wilcoxon signed-rank test. (**a**) Modified British Medical Research Council (mMRC) dyspnea index. (**b**) COPD assessment test (CAT) scores. (**c**) Maximal inspiratory pressure (PImax). (**d**) The 6 min walking distance (6MWD). * *p* < 0.05 through GEE, † *p* < 0.05, and ‡ *p* < 0.01 vs. T1.

**Figure 5 healthcare-11-02310-f005:**
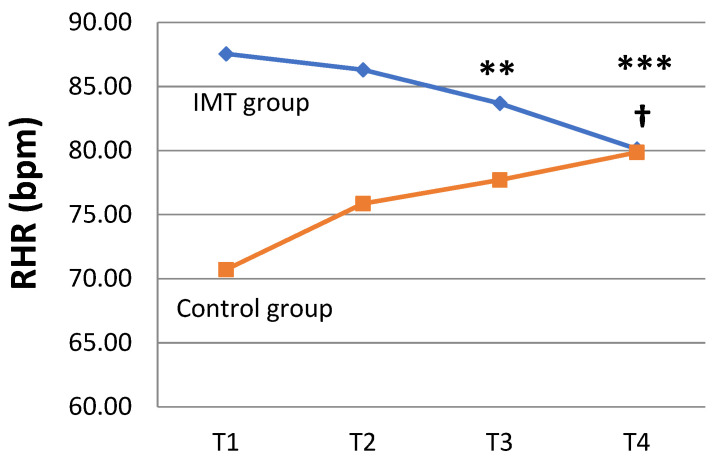
Changes in resting heart rate (RHR) from baseline to 12 weeks in the IMT and control groups. The interactions between two groups and the change between time points (from T1 to T2 (4 weeks), T3 (8 weeks), or T4) were compared and analyzed through GEE. ** *p* < 0.01 and *** *p* < 0.001. The differences in the IMT group at different time points vs. T1 were analyzed through Wilcoxon signed-rank test. † *p* < 0.05.

**Table 1 healthcare-11-02310-t001:** Participant demographics of the total study cohort and the two study groups.

Participant Demographics	Total Cohort (*n* = 23) N (%) or Mean ± SD	Study Groups		*p* Value
		IMT (*n* = 16) N (%) or Mean ± SD	Control (*n* = 7) N (%) or Mean ± SD	
Gender				0.273 ^a^
Male	19 (82.6)	12 (75.0)	7 (100.0)	
Female	4 (17.4)	4 (25.0)	0 (0.0)	
Age (years)	68.29 ± 10.87	67.25 ± 1.69	70.57 ± 12.66	0.535
BMI	23.54 ± 4.79	23.10 ± 4.26	24.55 ± 6.10	0.720
Smoking status				0.087 ^a^
Never smoker	10 (43.5)	9 (56.2)	1 (14.3)	
Former smoker	9 (39.1)	4 (25.0)	5 (71.4)	
Current smoker	4 (17.4)	3 (18.8)	1 (14.3)	
Severity of airflow limitation				0.202 ^a^
Moderate	15 (65.2)	12 (75.0)	3 (42.9)	
Severe	7 (30.4)	3 (18.8)	4 (57.1)	
Extremely severe	1 (4.3)	1 (6.3)	0 (0.0)	
Medication				
LAMA	1 (4.3)	1 (6.3)	0 (0.0)	1.000 ^a^
LABA	3 (13.0)	2 (12.5)	1 (14.3)	1.000 ^a^
LAMA/LABA	6 (26.1)	3 (18.8)	3 (42.9)	0.318 ^a^
ICS/LABA	6 (26.1)	3 (18.8)	3 (42.9)	0.318 ^a^
ICS/LAMA/LABA	6 (26.1)	6 (37.5)	0 (0.0)	0.124 ^a^
Theophylline	4 (17.4)	4 (25.0)	0 (0.0)	0.273 ^a^
OCS	4 (17.4)	4 (25.0)	0 (0.0)	0.273 ^a^
Number of AEs in the previous year	1.61 ± 1.95	2.00 ± 2.03	0.71 ± 1.50	0.154
≥2 AEs in the previous year	9 (39.1)	8 (50.0)	1 (14.3)	0.176 ^a^

Note: Data are shown as number of patients (%) for categorical variables and mean ± standard deviation for continuous variables. ^a^ Analyzed with Fisher’s exact test, as > 20% of the expected subjects had less than 5 in chi-square test. IMT represents inspiratory muscle training, BMI represents body mass index, LAMA represents long-acting muscarinic antagonist, LABA represents long-acting β2-agonist, ICS represents inhaled corticosteroid, OCS represents oral corticosteroid, and AE represents acute exacerbation.

**Table 2 healthcare-11-02310-t002:** Lung function and health status impairments in the two study groups.

Variable	Time	Study Group			
		IMT (*n* = 16) Mean ± SD	*p* Value	Control (*n* = 7) Mean ± SD	*p* Value
TLC (L)	T1	4.53 ± 1.01		6.48 ± 1.28	
	T4	4.90 ± 0.84	0.152	6.38 ± 1.11	0.612
RV (L)	T1	2.15 ± 0.87		4.02 ± 1.28	
	T4	2.18 ± 1.02	0.917	3.19 ± 1.69	0.553
RV/TLC%	T1	52.86 ± 11.06		61.00 ± 9.36	
	T4	49.77 ± 16.06	0.421	57.71 ± 10.09	0.916
FRC (L)	T1	3.07 ± 1.06		4.66 ± 1.15	
	T4	3.40 ± 0.85	0.064	4.60 ± 1.19	0.866
VC (L)	T1	2.09 ± 0.47		2.47 ± 0.55	
	T4	2.42 ± 0.73	0.011 *	2.63 ± 0.54	0.063
VT (L)	T1	0.73 ± 0.25		0.72 ± 0.17	
	T4	0.74 ± 0.20	0.484	0.81 ± 0.29	0.233
ERV (L)	T1	0.56 ± 0.38		0.48 ± 0.28	
	T4	0.90 ± 0.41	0.012 *	0.62 ± 0.36	0.345
DLCO	T1	9.80 ± 4.48		9.06 ± 4.11	
(mL/mmHg/min)	T4	9.98 ± 4.63	0.248	8.93 ± 3.66	0.933
DLCO/VA	T1	3.45 ± 1.16		2.42 ± 0.80	
(mL/mmHg/min/L)	T4	3.43 ± 1.20	1.000	2.24 ± 0.80	0.176
VA (L)	T1	2.90 ± 0.83		3.67 ± 1.25	
	T4	2.91 ± 0.70	0.133	4.02 ± 1.01	0.398
FVC (L)	T1	2.06 ± 0.50		2.32 ± 0.37	
	T4	2.10 ± 0.49	0.506	2.58 ± 0.61	0.063
FEV1 (L)	T1	1.16 ± 0.39		1.29 ± 0.37	
	T4	1.18 ± 0.36	0.442	1.26 ± 0.40	0.553
FEV1/FVC%	T1	55.71 ± 9.99		53.71 ± 11.70	
	T4	56.62 ± 9.35	0.916	49.43 ± 12.86	0.061
FEF_25–75%_ (L/s)	T1	0.57 ± 0.32		0.70 ± 0.33	
	T4	0.55 ± 0.22	0.889	0.48 ± 0.22	0.018 *
FEF_50%_ (L/s)	T1	0.74 ± 0.48		0.87 ± 0.51	
	T4	0.78 ± 0.45	0.552	0.65 ± 0.35	0.018 *
PEF (L/s)	T1	3.62 ± 1.47		3.95 ± 0.41	
	T4	3.60 ± 1.27	0.917	3.81 ± 0.94	0.499
mMRC (grade)	T1	1.50 ± 0.63		1.86 ± 0.90	
	T4	0.88 ± 0.62	0.008 *	1.86 ± 1.22	1.000
CAT (score)	T1	11.50 ± 6.16		11.29 ± 9.01	
	T4	8.88 ± 6.54	0.064	12.57 ± 10.47	0.395
HR (bpm)	T1	87.56 ± 13.04		70.71 ± 12.83	
	T4	80.13 ± 10.23	0.011 *	79.86 ± 10.42	0.028 *
PI_max_ (cmH_2_O)	T1	72.13 ± 26.75		88.57 ± 26.07	
	T4	84.31 ± 23.72	0.011 *	91.43 ± 27.46	1.000
6MWD (m)	T1	424.44 ± 66.40		364.00 ± 130.90	
	T4	442.38 ± 68.01	0.070	406.57 ± 82.43	0.204

Note: * *p* < 0.05. IMT represents inspiratory muscle training, T1 represents baseline, T4 represents 12 weeks, TLC represents total lung capacity, RV represents residual volume, FRC represents functional residual capacity, VC represents vital capacity, VT represents tidal volume, ERV represents expiratory reserve volume, DLCO represents diffusion capacity of the lung for carbon monoxide, VA represents alveolar ventilation, FVC represents forced vital capacity, FEV_1_ represents forced expiratory volume in one second, FEF_25–75%_ represents forced expiratory flow between 25 and 75% of the FVC, FEF_50%_ represents forced expiratory flow at 50% of the FVC, PEF represents peak expiratory flow, mMRC represents modified Medical Research Council dyspnea index, CAT represents COPD Assessment Test, HR represents heart rate, bpm represents beats per minute, PI_max_ represents maximal inspiratory pressure, and 6MWD represents 6 min walking distance.

**Table 3 healthcare-11-02310-t003:** Analysis of the effects of interventional IMT on the lung function and health status impairments in COPD participants.

Study Variable		Beta Coefficient (*B*)	95% Wald Confidence Interval			*p*-Value
FEV1/FVC%	Intercept	53.71	45.69	to	61.74	0.000 ^ǂ^
	Study group (IMT vs. control)	2.00	−7.48	to	11.48	0.679
	Time point					
	T4 vs. T1	−4.29	−7.24	to	−1.33	0.004 ^ƚ^
	Study group × time point					
	IMT × (T4–T1)	5.21	0.46	to	9.96	0.032 *
FEF_25–75%_	Intercept	0.70	0.47	to	0.92	0.000 ^ǂ^
(L/s)	Study group (IMT vs. control)	−0.13	−0.41	to	0.15	0.370
	Time point					
	T4 vs. T1	−0.22	−0.32	to	−0.12	0.000 ^ǂ^
	Study group × time point					
	IMT × (T4–T1)	0.20	0.04	to	0.35	0.012 *
FEF_50%_	Intercept	0.87	0.52	to	1.22	0.000 ^ǂ^
(L/s)	Study group (IMT vs. control)	−0.13	−0.56	to	0.29	0.546
	Time point					
	T4 vs. T1	−0.22	−0.36	to	−0.09	0.000 ^ǂ^
	Study group × time point					
	IMT × (T4–T1)	0.26	0.08	to	0.43	0.004 ^ƚ^
mMRC	Intercept	1.86	1.24	to	2.47	0.000 ^ǂ^
(grade)	Study group (IMT vs. control)	−0.36	−1.04	to	0.33	0.308
	Time point					
	T2 vs. T1	−0.29	−1.23	to	0.66	0.554
	T3 vs. T1	−0.29	−0.94	to	0.37	0.391
	T4 vs. T1	−3.37 × 10^−17^	−0.56	to	0.56	1.000
	Study group × time point					
	IMT × (T2–T1)	−0.03	−1.02	to	0.96	0.958
	IMT × (T3–T1)	−0.21	−0.91	to	0.48	0.547
	IMT × (T4–T1)	−0.63	−1.28	to	0.03	0.062
CAT	Intercept	11.29	5.10	to	17.47	0.000 ^ǂ^
(score)	Study group (IMT vs. control)	0.21	-6.62	to	7.05	0.951
	Time point					
	T2 vs. T1	2.29	-1.94	to	6.51	0.289
	T3 vs. T1	3.00	-1.37	to	7.37	0.179
	T4 vs. T1	1.29	-3.69	to	6.26	0.612
	Study group × time point					
	IMT × (T2–T1)	−4.54	−9.47	to	1.61	0.071
	IMT × (T3–T1)	−5.50	−10.31	to	−0.695	0.025 *
	IMT × (T4–T1)	−3.91	−9.42	to	1.61	0.165
RHR (bpm)	Intercept	70.71	61.92	to	79.51	0.000 ^ǂ^
	Study group (IMT vs. control)	16.85	6.09	to	27.60	0.002 ^ƚ^
	Time point					
	T2 vs. T1	5.14	−0.57	to	10.34	0.053
	T3 vs. T1	7.00	2.37	to	11.64	0.003 ^ƚ^
	T4 vs. T1	9.14	3.82	to	14.46	0.001 ^ƚ^
	Study group × time point					
	IMT × (T2–T1)	−6.39	−13.29	to	0.51	0.069
	IMT × (T3–T1)	−10.88	−18.22	to	−3.53	0.004 ^ƚ^
	IMT × (T4–T1)	−16.58	−23.51	to	−9.65	0.000 ^ǂ^

Note: * *p* < 0.05, ^ƚ^
*p* < 0.01, and ^ǂ^
*p* < 0.001. FVC represents forced vital capacity, FEV_1_ represents forced expiratory volume in one second, FEF_25–75%_ represents forced expiratory flow between 25 and 75% of the FVC, FEF_50%_ represents forced expiratory flow at 50% of the FVC, mMRC represents modified Medical Research Council dyspnea index, CAT represents COPD assessment test, RHR represents resting heart rate, and bpm represents beats per minute.

## Data Availability

Any data related to this study can be provided upon a reasonable request.

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
