# Peer review of "The Effectiveness of Home-Based Inspiratory Muscle Training on Small Airway Function and Disease-Associated Symptoms in Patients with Chronic Obstructive Pulmonary Disease"

_healthcare, 2023, doi:10.3390/healthcare11162310_

Round 1
Reviewer 1 Report
In this study, the authors used the IMT device to improve pulmonary function in patients with COPD, which has research contribution in this field, but there are still some shortcomings:
Induction
1. Inaccurate text presentation, for example, different statements on " COPD is one of the top three leading causes of deaths" and "COPD is ranked 8th in the top-ten leading causes of death" in Introduction part.
Method
1. “a diagnosis of COPD with post-bronchodilator forced vital capacity (FEV1/FVC) <0.7 and at least moderate airflow limitation according to the 2011 Global Initiative for Chronic Obstructive Lung Disease (GOLD) criteria”,please add references here.
2. Please add the reasons and references for the selection of these assessment metrics in section 2.3.
3. Why 40% was used as the threshold in 2.3.1.
4. Please refer to the standard protocol for respiratory muscle measurements in 2.3.2.
5. Please describe in detail the care of the control group in the method.
6. Please add sample size estimation in the methods.
7. Whether patient medication was controlled during the intervention?
8. Please add the detailed procedure of pulmonary function measurement to the method.
Results
1. Although the inclusion criteria for this study in the methodology was over 40 years, the mean age of participants in the actual results was over 67 years, suggesting that the title and questions focus on the improvement of older patients with COPD.
2. There was a large difference in the gender of participants, with fewer women, so please interpret the results with caution.
Conclusions
1. From a sample size perspective, the study has more limitations, so please be careful in your conclusions.
Author Response
|
Reviewer 1 |
Reply |
|
In this study, the authors used the IMT device to improve pulmonary function in patients with COPD, which has research contribution in this field, but there are still some shortcomings: Induction 1. Inaccurate text presentation, for example, different statements on " COPD is one of the top three leading causes of deaths" and "COPD is ranked 8th in the top-ten leading causes of death" in Introduction part. |
1.Revised and added leading causes of deaths in “Taiwan” in Line 77-80.
|
|
Method 1. “a diagnosis of COPD with post-bronchodilator forced vital capacity (FEV1/FVC) <0.7 and at least moderate airflow limitation according to the 2011 Global Initiative for Chronic Obstructive Lung Disease (GOLD) criteria”,please add references here. 2. Please add the reasons and references for the selection of these assessment metrics in section 2.3. 3. Why 40% was used as the threshold in 2.3.1. 4. Please refer to the standard protocol for respiratory muscle measurements in 2.3.2. 5. Please describe in detail the care of the control group in the method. 6. Please add sample size estimation in the methods. 7. Whether patient medication was controlled during the intervention? 8. Please add the detailed procedure of pulmonary function measurement to the method. |
1. Added references in Line107-112.
2. Added reasons and references for these assessment metrics in Line 121-125. 3. Added and more explained in Line 67-71, 145-148. 4. Added in Line 182-191.
5. Added and more details explained in Line 83-86, 104-107. 6. Revised in Line 465-473. 7. Yes. Added and more explained in Line 83-86, 104-107, 113-114. 8. Added in Line 170-175. |
|
Results 1. Although the inclusion criteria for this study in the methodology was over 40 years, the mean age of participants in the actual results was over 67 years, suggesting that the title and questions focus on the improvement of older patients with COPD. 2. There was a large difference in the gender of participants, with fewer women, so please interpret the results with caution. |
1. Added and explained more details in Discussion section, in Line 455-458.
2. Revised and added more details in Line 473-478. |
|
Conclusions 1. From a sample size perspective, the study has more limitations, so please be careful in your conclusions. |
1.Revised in Line 480-483. |
Reviewer 2 Report
Dear Authors,
Manuscript ID: healthcare-2502201
Title Manuscript: The Effectiveness of Home-based Inspiratory Muscle Training on Small Airway Function and Disease-associated Symptoms in Patients with Chronic Obstructive Pulmonary Disease
This non-randomized controlled study examined the effectiveness of 12 weeks of home-based inspiratory muscle training on small airway function and disease-associated symptoms in patients with chronic obstructive pulmonary disease (COPD). This is an interesting topic since the study participants are patients with COPD but at the moment MAJOR REVISIONS are necessary in order to make it suitable for a final decision for “Healthcare”.
POINTs of STRENGTH:
1) Effects of 12 weeks of home-based inspiratory muscle training on small airway function and disease-associated symptoms in patients with COPD;
2) Arguments in the discussion section;
POINTs of WEAKNESS (and/or should be revised to improve the manuscript):
Abstract:
3) The gender, number of participants in each group, mean age, BMI and the type of exercise are not specify. Please specify in the methods section of the Abstract;
4) The significance level in the results section is unclear. Please specify;
5) Please add the keyword “Chronic obstructive pulmonary disease” in the keywords section;
1. Introduction:
6) The hypothesis and purposes of this study are not specify. Please specify clearly;
2. Materials and Methods
2.1. Study design & 2.2. Participants
7) The recruitment process of participants OR inclusion and exclusion criteria should be described in more detail such as age, gender, BMI, metabolic disorders, physical fitness level/VO2max, METs, and so on;
8) The control group can be stated in more detail. Please provide and add;
2.3.1. The IMT device and prescribed training
9) The type of training/exercise is unclear. Please specify;
10) Considering the important effects of nutrition on lung function and other parameters, did the authors monitor the nutritional status of participants (patients with COPD) during 12 weeks? IF YES, please add the method of controlling nutritional status with Pre- and Post-data;
2.4. Statistical analysis
11) It seems that the total number of participants for this study is small. Did authors use a statistical software to calculate the sample size? IF YES, please add software name and its findings in the Statistical analysis section;
12) The significance level of statistical analysis was considered for one-tailed? OR two-tailed? Please specify;
4. Discussion & 5. Conclusions
13) Discussion section is well-written;
14) As mentioned above, the authors will agree that the limitation section has to be expanded. In addition, the authors did not give any points of strength to this study, which I encourage;
15) Please provide clinical perspectives for patients with COPD OR lung disorders;
16) What does this study add to the literature for exercise intervention-related studies in patients with COPD? Please explain.
References
17) “References” section is not always in accordance with the authors' guidelines. In particular, please check No. 1, 13, 17, 19, 21, 22, 26, 31, and 36 for validation.
Best Regards
11 July 2023
Author Response
|
Reviewer 2 |
Reply |
|
This non-randomized controlled study examined the effectiveness of 12 weeks of home-based inspiratory muscle training on small airway function and disease-associated symptoms in patients with chronic obstructive pulmonary disease (COPD). This is an interesting topic since the study participants are patients with COPD but at the moment MAJOR REVISIONS are necessary in order to make it suitable for a final decision for “Healthcare”. POINTs of STRENGTH: 1) Effects of 12 weeks of home-based inspiratory muscle training on small airway function and disease-associated symptoms in patients with COPD; 2) Arguments in the discussion section; POINTs of WEAKNESS (and/or should be revised to improve the manuscript): Abstract: 3) The gender, number of participants in each group, mean age, BMI and the type of exercise are not specify. Please specify in the methods section of the Abstract; 4) The significance level in the results section is unclear. Please specify; 5) Please add the keyword “Chronic obstructive pulmonary disease” in the keywords section; |
1) & 2) Thanks.
3)Added in Line24-26
4) Revised in Line27-31. 5) Added in Line 35.
|
|
1. Introduction: 6) The hypothesis and purposes of this study are not specify. Please specify clearly;
|
6)Added in Line 89-93. |
|
2. Materials and Methods 2.1. Study design & 2.2. Participants 7) The recruitment process of participants OR inclusion and exclusion criteria should be described in more detail such as age, gender, BMI, metabolic disorders, physical fitness level/VO2max, METs, and so on; 8) The control group can be stated in more detail. Please provide and add; 2.3.1. The IMT device and prescribed training 9) The type of training/exercise is unclear. Please specify; 10) Considering the important effects of nutrition on lung function and other parameters, did the authors monitor the nutritional status of participants (patients with COPD) during 12 weeks? IF YES, please add the method of controlling nutritional status with Pre- and Post-data; 2.4. Statistical analysis 11) It seems that the total number of participants for this study is small. Did authors use a statistical software to calculate the sample size? IF YES, please add software name and its findings in the Statistical analysis section; 12) The significance level of statistical analysis was considered for one-tailed? OR two-tailed? Please specify; |
7)Revised and explained in Line 107-119.
8) Added and more details explained in Line 83-86, 104-107.
9) Revised and added in Line 65-71, 145-148. 10) We did not monitor the nutritional status of participants during 12 weeks. We make recommendations for future research in Line 455-464.
11) Added and more details explained of the sample size in Line 465-473.
12) Added in Line 224. |
|
4. Discussion & 5. Conclusions 13) Discussion section is well-written; 14) As mentioned above, the authors will agree that the limitation section has to be expanded. In addition, the authors did not give any points of strength to this study, which I encourage; 15) Please provide clinical perspectives for patients with COPD OR lung disorders; 16) What does this study add to the literature for exercise intervention-related studies in patients with COPD? Please explain. |
13) Thanks.
14) Revised and added in Line 447-454.
15) Revised and added in Line 442-445. 16) Revised and added more information in Line 65-71, 455-464. |
|
References 17) “References” section is not always in accordance with the authors' guidelines. In particular, please check No. 1, 13, 17, 19, 21, 22, 26, 31, and 36 for validation.
|
17) Revised of the References. |
Reviewer 3 Report
Dear Authors,
Strong evidence in literature currently depicts COPD as a condition with high mortality and disability rates. In this scenario, developing effective rehabilitation strategies is a crucial point in lowering comorbidities and improving the patients’ quality of life. In light of these considerations, I think that the present manuscript could offer an intriguing perspective paving the way for future research. However, reviewing the present manuscript rises some critical concerns, as described below.
Major reviews
INTRODUCTION: lines 50-59. In my opinion, the section is well-focused on the goals of pulmonary rehabilitation in COPD patients. However, further insights on the complex management of osteosarcopenia should be provided. In addition, as your manuscript assessed a home-based training program, I think that the benefits of telerehabilitation in an outpatient setting should be discussed in further detail within this section. You might consider citing the following literature:
- Lippi L, Folli A, Curci C, D'Abrosca F, Moalli S, Mezian K, de Sire A, Invernizzi M. Osteosarcopenia in Patients with Chronic Obstructive Pulmonary Diseases: Which Pathophysiologic Implications for Rehabilitation? Int J Environ Res Public Health. 2022 Nov 2;19(21):14314. doi: 10.3390/ijerph192114314.
- Lippi L, Turco A, Folli A, D'Abrosca F, Curci C, Mezian K, de Sire A, Invernizzi M. Technological advances and digital solutions to improve quality of life in older adults with chronic obstructive pulmonary disease: a systematic review. Aging Clin Exp Res. 2023 May;35(5):953-968. doi: 10.1007/s40520-023-02381-3.
- Vilarinho R, Serra L, Coxo R, Carvalho J, Esteves C, Montes AM, Caneiras C. Effects of a Home-Based Pulmonary Rehabilitation Program in Patients with Chronic Obstructive Pulmonary Disease in GOLD B Group: A Pilot Study. Healthcare (Basel). 2021 May 4;9(5):538. doi: 10.3390/healthcare9050538.
METHODS: apparently, some information about the sample size calculation is missing. This aspect is crucial for ensuring the internal validity and reliability of the study results.
DISCUSSION: please, note that the study sample is limited to a specific geographic region and a single medical center, which raises concerns about the generalizability of the findings to a broader population of COPD patients. Indeed, it is important to consider the potential impact of regional variations in COPD prevalence, healthcare practices, and cultural factors that may influence the outcomes. In addition, the study design is non-randomized and quasi-experimental, which may introduce bias and confounding factors that could influence the outcomes. It is essential to address potential sources of bias adequately, such as participant selection bias. I think that the Discussion section should be implemented keeping in mind the aforementioned consideration, providing a small discussion of this manuscript’s limitations.
Given these concerns, I recommend that you address these issues and provide additional details and clarification in the complete manuscript. I appreciate your attention and I trust that you will thoroughly evaluate and revise the manuscript to address these issues and improve its scientific validity and contribution to the field.
Thank you for your time and consideration.
Best regards
Author Response
|
Reviewer 3 |
|
|
Major reviews INTRODUCTION: lines 50-59. In my opinion, the section is well-focused on the goals of pulmonary rehabilitation in COPD patients. However, further insights on the complex management of osteosarcopenia should be provided. In addition, as your manuscript assessed a home-based training program, I think that the benefits of telerehabilitation in an outpatient setting should be discussed in further detail within this section. You might consider citing the following literature: - Lippi L, Folli A, Curci C, D'Abrosca F, Moalli S, Mezian K, de Sire A, Invernizzi M. Osteosarcopenia in Patients with Chronic Obstructive Pulmonary Diseases: Which Pathophysiologic Implications for Rehabilitation? Int J Environ Res Public Health. 2022 Nov 2;19(21):14314. doi: 10.3390/ijerph192114314. - Lippi L, Turco A, Folli A, D'Abrosca F, Curci C, Mezian K, de Sire A, Invernizzi M. Technological advances and digital solutions to improve quality of life in older adults with chronic obstructive pulmonary disease: a systematic review. Aging Clin Exp Res. 2023 May;35(5):953-968. doi: 10.1007/s40520-023-02381-3. - Vilarinho R, Serra L, Coxo R, Carvalho J, Esteves C, Montes AM, Caneiras C. Effects of a Home-Based Pulmonary Rehabilitation Program in Patients with Chronic Obstructive Pulmonary Disease in GOLD B Group: A Pilot Study. Healthcare (Basel). 2021 May 4;9(5):538. doi: 10.3390/healthcare9050538. |
Revised and added more information related sarcopenia in Line 458-464 and references No. 41 (Lippi L, Folli A, Curci C, D'Abrosca F, Moalli S, Mezian K, de Sire A, Invernizzi M. Osteosarcopenia in Patients with Chronic Obstructive Pulmonary Diseases: Which Pathophysiologic Implications for Rehabilitation? Int J Environ Res Public Health. 2022 Nov 2;19(21):14314.
|
|
METHODS: apparently, some information about the sample size calculation is missing. This aspect is crucial for ensuring the internal validity and reliability of the study results.
|
Revised and added in Line 465-473. |
|
DISCUSSION: please, note that the study sample is limited to a specific geographic region and a single medical center, which raises concerns about the generalizability of the findings to a broader population of COPD patients. Indeed, it is important to consider the potential impact of regional variations in COPD prevalence, healthcare practices, and cultural factors that may influence the outcomes. In addition, the study design is non-randomized and quasi-experimental, which may introduce bias and confounding factors that could influence the outcomes. It is essential to address potential sources of bias adequately, such as participant selection bias. I think that the Discussion section should be implemented keeping in mind the aforementioned consideration, providing a small discussion of this manuscript’s limitations.
|
Revised and added in Line 455-464, 473-478, and 480-483. |
Round 2
Reviewer 1 Report
Looks better. No more questions.
Reviewer 2 Report
Dear Authors,
Manuscript ID: healthcare-2502201
Title Manuscript: The Effectiveness of Home-based Inspiratory Muscle Training on Small Airway Function and Disease-associated Symptoms in Patients with Chronic Obstructive Pulmonary Disease
I am very grateful to the authors for their efforts.
In general, this manuscript has found suitable content after correcting major revisions, and the modified revisions are accepted.
Best Regards
1 August 2023
Reviewer 3 Report
Dear Authors,
I have assessed the latest version of your manuscript. In my opinion, the overall quality of the paper has significantly improved after the newly added methodological insights and valuable comments. In light of this consideration, I have no further suggestions, as I think that the present version of the paper would be suitable for publication.
Best regards,